# Neural Policy Ensembles are Sub-Optimal

## Abstract

We develop a theoretical framework to formally prove that (non-linear) neural policy ensembles are sub-optimal with respect to linear policy ensembles. We empirically validate our theoretical claims through a variety of comparisons between policy ensembles composed of linear and of (non-linear) neural policies. We empirically show that well-tuned neural policy ensembles $\Pi^N$ under-perform equivalent linear ensembles, often by 2 orders of magnitude. We further show that, under identical operating conditions for ensembles of policies (each of which is stable), $\Pi^N$ can show significant instability while linear policy ensembles are stable. This sub-optimality has significant implications for all neural policy ensemble research, from those based on Reinforcement Learning to Mixture-of-Expert agentic-AI policies.

## 1 Introduction

Ensemble methods are cornerstone techniques in machine learning, with strong theoretical foundations showing that diversity among classifiers leads to improved performance through bias-variance reduction (Yang et al., 2023). Recently, researchers have applied ensemble techniques to policy ensemble methods, particularly in reinforcement learning (RL), where they combine multiple policies to improve generalization, robustness, and sample efficiency (Yang et al., 2022a; Wiering & van Hasselt, 2008). These methods can be homogeneous (same algorithm with different seeds) or heterogeneous (different algorithms or architectures), and may be meta-learned to adapt dynamically. In addition to this work, Mixture-of-Experts (MoE) architectures, which have gained prominence in agentic AI and large language model (LLM) applications, combine multiple specialized policies or models to improve scalability, adaptability, and performance across diverse tasks, e.g., (Liu, 2025; Willi et al., 2024; Celik et al., 2024b).

We show that a weighted ensemble $\Pi$ of nonlinear *policies* (implemented by approaches like neural networks) is *inherently sub-optimal* compared to the individual policies in $\Pi$ or to ensembles of linear policies, when both neural and linear ensembles are trained from identical data. We develop a theoretical framework to formally prove this sub-optimality by analyzing the fundamental differences between ensemble classifiers and ensemble policies from a temporal function approximation perspective. We empirically validate our theoretical background through a variety of comparisons between linear and (non-linear) neural ensembles. We show that well-tuned neural ensembles underperform in comparison to equivalent linear ensembles, often by 2 orders of magnitude.

The intuitive difference between ensemble classifiers and ensemble policies is as follows. Ensemble classifiers benefit from independence: individual errors cancel through averaging because each classifier sees independent samples from a fixed distribution. In contrast, nonlinear policy ensembles face temporal coupling: the ensemble's actions affect future states, creating feedback loops that may amplify rather than cancel errors. The temporal dependence breaks the mathematical foundation of ensemble methods.

This suggests a fundamental limitation: *nonlinear function approximators are inherently unsuitable for ensemble control methods*, regardless of how sophisticated the ensemble design becomes. The classical adaptive control theory that works for linear/parametric controllers (Kuipers & Ioannou, 2010) doesn't extend to neural networks. As a consequence, work in RL and LLM MoE settings that employ neural policy ensembles need to carefully consider how they adopt this approach. In addition, agentic AI may need to carefully examine its functionality to ensure that policy MoE approaches are designed to adhere to the optimality principles introduced in this article.

### 1.1 Contributions

- **Sub-Optimality:** We prove that an ensemble neural network policy will perform sub-optimally compared to individual policies, while an ensemble linear policy maintains optimality guarantees.

- **Stability:** We prove that a neural ensemble policy does not guarantee stability even when all individual policies are stable, whereas a linear policy ensemble composed of stable linear policies guarantees stability; these results hold for varying rates of nonstationary change.

- **Neural Mixing:** We prove that, even if you have a set of optimal linear policies, using a neural network to mix the policies is sub-optimal.

- **Empirical validation:** We experimentally validate our theoretical claims, using (1) optimality experiments based on both linear and non-linear dynamical systems, and (2) stability experiments on linear systems with varying rates of nonstationarity.

## 2 MATHEMATICAL FRAMEWORK

### 2.1 SYSTEM MODEL AND ASSUMPTIONS

Consider a continuous-time nonlinear dynamical system $\dot{x}(t) = f(x(t), u(t)) + w(t)$, where $x(t) \in \mathbb{R}^n$ is the state, $u(t) \in \mathbb{R}^m$ is the control input, $f : \mathbb{R}^n \times \mathbb{R}^m \to \mathbb{R}^n$ is Lipschitz continuous with constant $L_f$, and $w(t)$ represents bounded disturbances.

**Definition 1** (Admissible Policies). *A policy $\pi : \mathbb{R}^n \to \mathbb{R}^m$ is admissible if:*

1. *$\pi$ is measurable and locally Lipschitz;*

2. *The closed-loop system $\dot{x} = f(x, \pi(x)) + w$ has unique solutions;*

3. *$\pi$ stabilizes the origin in the absence of disturbances.*

### 2.2 OPTIMAL CONTROL FORMULATION

**Definition 2** (Value Function). *For an admissible policy $\pi$ and initial state $x_0$, define the infinite-horizon discounted value function:*

$$V^\pi(x_0) = \mathbb{E}\left[\int_0^\infty e^{-\rho t} \ell(x(t), \pi(x(t)))\, dt \,\Big|\, x(0) = x_0\right] \tag{1}$$

*where $\rho > 0$ is the discount rate and $\ell(x, u) = x^T Q x + u^T R u$ with $Q \succeq 0$, $R \succ 0$.*

**Definition 3** (Optimal Value Function). *The optimal value function is:*

$$V^*(x) = \inf_{\pi \in \tilde{\Pi}} V^\pi(x), \tag{2}$$

*where $\tilde{\Pi}$ denotes the set of admissible policies.*

**Definition 4** (Hamilton-Jacobi-Bellman Equation). *The optimal value function satisfies the HJB equation:*

$$\rho V^*(x) = \min_{u \in \mathbb{R}^m} \left\{ \ell(x, u) + \nabla V^*(x)^T f(x, u) \right\} \tag{3}$$

*with optimal policy $\pi^*(x) = \mathrm{argmin}_u \{\ell(x, u) + \nabla V^*(x)^T f(x, u)\}$.*

### 2.3 POLICY ENSEMBLES IN OPTIMAL CONTROL

#### 2.3.1 LINEAR POLICY ENSEMBLES

**Definition 5** (Linear Policy). *A linear policy has the form $\pi^L(x) = Kx$ where $K \in \mathbb{R}^{m \times n}$.*

**Definition 6** (Linear Policy Ensemble). *Given linear policies $\{\pi_i^L(x) = K_i x\}_{i=1}^M$ and weights $w_i \geq 0$ with $\sum_{i=1}^M w_i = 1$, the linear ensemble is:*

$$\Pi^L(x) = \sum_{i=1}^M w_i K_i x = K_{ens} x \qquad \text{where } K_{ens} = \sum_{i=1}^M w_i K_i. \tag{4}$$

### 2.3.2 NEURAL POLICY ENSEMBLES

**Definition 7** (Neural Network Policy). *A neural network policy $\pi^\theta : \mathbb{R}^n \to \mathbb{R}^m$ is parameterized by weights $\theta$ and has the form:*

$$\pi^\theta(x) = W_L \sigma(W_{L-1} \sigma(\cdots \sigma(W_1 x + b_1) \cdots) + b_{L-1}) + b_L \tag{5}$$

*where $\sigma$ is a nonlinear activation function and $\{W_i, b_i\}$ are the network parameters.*

**Definition 8** (Neural Policy Ensemble). *Given neural policies $\{\pi^{\theta_i}\}_{i=1}^M$ and weights $w_i \geq 0$ with $\sum_{i=1}^M w_i = 1$, the neural ensemble is:*

$$\Pi^N(x) = \sum_{i=1}^M w_i \pi^{\theta_i}(x) \tag{6}$$

## 3 MAIN THEORETICAL RESULTS

### 3.1 SUBOPTIMALITY ANALYSIS FOR NEURAL POLICIES

**Definition 9** (Policy Suboptimality). *For any policy $\pi$, define the suboptimality gap as:*

$$\Delta(\pi, x) = V^\pi(x) - V^*(x) \geq 0 \tag{7}$$

**Definition 10** (Nonlinearity Measure). *For a neural policy $\pi^\theta$ on a domain $D \subset \mathbb{R}^n$, define:*

$$\kappa(\pi^\theta, D) = \sup_{\substack{x,y \in D \\ x \neq y}} \sup_{\lambda \in [0,1]} \frac{\|\pi^\theta(\lambda x + (1-\lambda)y) - \lambda \pi^\theta(x) - (1-\lambda)\pi^\theta(y)\|}{\|x - y\|} \tag{8}$$

Our main result about policy ensemble sub-optimality is as follows:

**Theorem 1** (Neural Ensemble Suboptimality). *Consider a stabilizable linear system $\dot{x} = Ax + Bu$ with neural policies $\{\pi^{\theta_i}\}_{i=1}^M$ and corresponding optimal linear policies $\{\pi_i^L = K_i^* x\}_{i=1}^M$ solving individual LQR problems with cost matrices $(Q_i, R_i)$.*

*Let $D = \{x \in \mathbb{R}^n : \|x\| \leq R\}$ be a bounded operating region. If:*

1.  ***Diversity:*** $\min_{i \neq j} \|K_i^* - K_j^*\|_F \geq \delta > 0$

2.  ***Nonlinearity:*** $\min_i \kappa(\pi^{\theta_i}, D) \geq \kappa_0 > 0$

3.  ***Sufficient Complexity:*** $L_f \kappa_0 \delta > \rho$

*Then there exists $\epsilon(\kappa_0, \delta, L_f) > 0$ such that: $\sup_{x \in D} \left[ V^{\Pi^N}(x) - V^{\Pi^L}(x) \right] \geq \epsilon(\kappa_0, \delta, L_f)$.*

### 3.2 STABILITY ANALYSIS VIA CONTROL LYAPUNOV FUNCTIONS

**Definition 11** (Control Lyapunov Function). *A smooth function $V : \mathbb{R}^n \to \mathbb{R}_+$ is a Control Lyapunov Function (CLF) for system $\dot{x} = f(x, u)$ if:*

1.  $V(0) = 0$ *and* $V(x) > 0$ *for* $x \neq 0$

2.  $\inf_{u \in \mathbb{R}^m} \{\nabla V(x)^T f(x, u)\} < 0$ *for all* $x \neq 0$

Our main result about policy ensemble stability is as follows:

**Theorem 2** (Stability Violation in Neural Ensembles). *Consider neural policies $\{\pi^{\theta_i}\}_{i=1}^M$, each with CLF $V_i(x)$ satisfying:*

$$\nabla V_i(x)^T f(x, \pi^{\theta_i}(x)) \leq -\alpha_i \|x\|^2 \text{ for some } \alpha_i > 0. \tag{9}$$

*If the ensemble weights vary as $w_i(t)$ with $\|\dot{w}(t)\| \geq \beta > 0$, then the ensemble system can be unstable even when all individual systems are stable. Specifically, if $\beta > \frac{\min_i \alpha_i}{2 \max_i \|V_i\|_\infty}$, then there exist initial conditions where the ensemble trajectory is unbounded.*

## 3.3 Theoretical Analysis of Policy Mixing

We now examine the impact of using nonlinear algorithms for policy mixing, for the cases where the underlying policy ensemble consists of linear and of nonlinear policies. Several works use neural networks for mixing policies, e.g., (Burke et al., 2020; Kim et al., 2000; Jacobs et al., 1991). We now show the sub-optimality of this approach, with respect to convex mixing.

**Definition 12** (Policy Mixing). *Given $N$ policies $\pi_i : \mathcal{X} \to \mathcal{U}$ and weights $w \in \mathbb{R}^N$, the ensemble policy is:* $\Pi(x) = \sum_{i=1}^N w_i \pi_i(x)$. *We distinguish:*

- ***Convex mixing**: $w_i \geq 0, \sum w_i = 1$*

- ***Non-convex mixing**: $w \in \mathbb{R}^N \setminus \Delta^{N-1}$ where $\Delta^{N-1}$ is the probability simplex*

**Definition 13** (Performance Measure). *For a cost function $J$ and initial state distribution $\mu_0$, define:*

$$\mathcal{L}(w) = \mathbb{E}_{x_0 \sim \mu_0} \left[ J(x_0, \pi_{ens}(x_0)) \right]$$

We now show the entailed properties for Linear Quadratic Systems. We can show that Lemma 2 holds for a system with linear time-invariant dynamics and quadratic costs, which is defined as follows:

$$x_{t+1} = Ax_t + Bu_t \tag{10}$$

$$J_i(x, u) = x^T Q_i x + u^T R_i u, \quad i = 1, \ldots, N, \tag{11}$$

where $Q_i \succeq 0$, $R_i \succ 0$, and $(A, B)$ is stabilizable.

**Lemma 1** (Optimal Linear Policies). *For each regime $i$, the optimal controller is linear: $\pi_i^*(x) = -K_i x$ where*

$$K_i = (R_i + B^T P_i B)^{-1} B^T P_i A$$

*and $P_i$ solves the algebraic Riccati equation:*

$$P_i = Q_i + A^T P_i A - A^T P_i B (R_i + B^T P_i B)^{-1} B^T P_i A$$

### 3.3.1 Main Theoretical Result (nonconvex mixing)

Our main result states that nonconvex (e.g., neural) mixing is sub-optimal:

**Theorem 3** (Convexity Advantage for Weighted Average Cost). *Let $\lambda \in \Delta^{N-1}$ be a probability vector, and define the weighted average cost:*

$$J_\lambda(x, u) = \sum_{i=1}^N \lambda_i J_i(x, u) = x^T Q_\lambda x + u^T R_\lambda u$$

*where $Q_\lambda = \sum \lambda_i Q_i$ and $R_\lambda = \sum \lambda_i R_i$.*

*Then for any mixing weights $w \in \mathbb{R}^N$:*

$$\mathcal{L}_\lambda(w) \geq \mathcal{L}_\lambda(\lambda)$$

*with equality if and only if $w = \lambda$.*

An immediate corollary is as follows:

**Corollary 1** (Performance Bound). *The performance penalty for non-convex mixing is:*

$$\mathcal{L}_\lambda(w) - \mathcal{L}_\lambda(\lambda) = \mathbb{E}[x_0^T (K_w - K_\lambda)^T R_\lambda (K_w - K_\lambda) x_0] \geq 0$$

## 4 Empirical Study of Performance

We have conducted multiple empirical studies to investigate the performance gap between neural network (NN) ensembles and linear quadratic regulator (LQR) ensembles in a multi-regime control environment. The goal is to analyze how convexity violations in NN-based controllers affect performance compared to theoretically optimal LQR controllers.

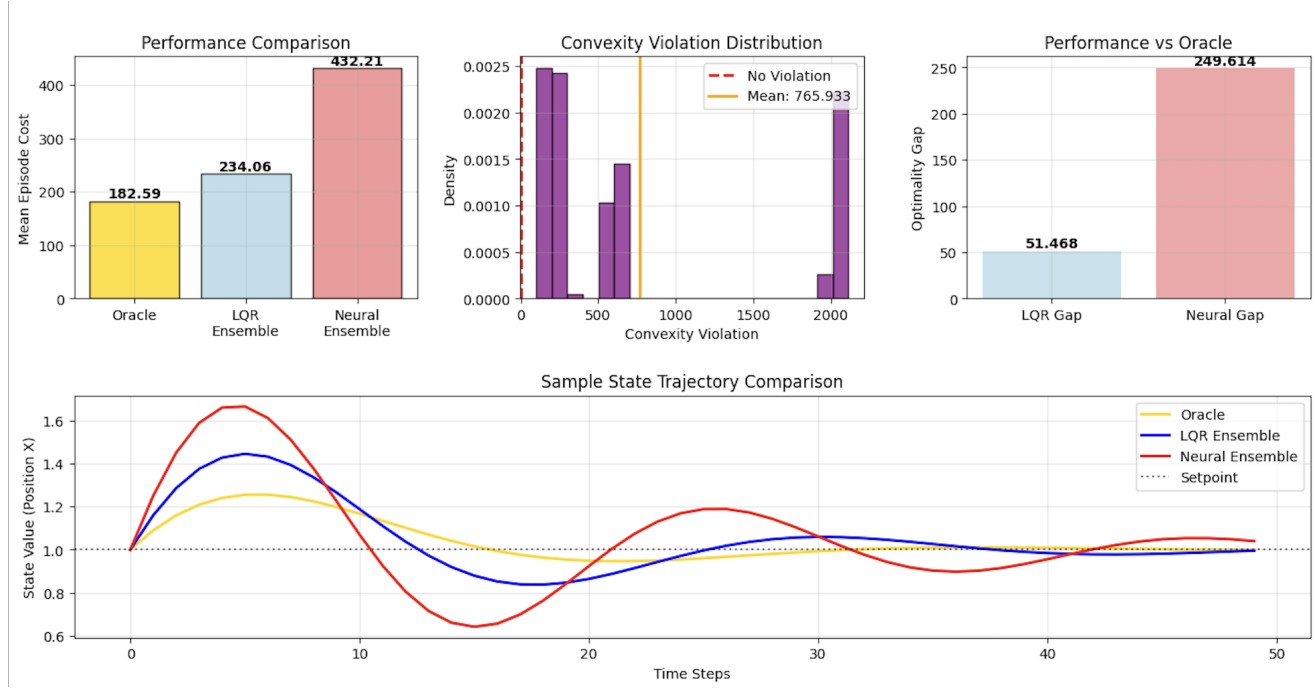

Figure 1: Experimental results for Multi-Regime Linear Dynamical System

## 4.1 LINEAR DYNAMICAL SYSTEM

This implementation provides a well-conditioned linear dynamical system with multiple operating regimes (each with different objectives) for ensemble control analysis. It is designed to be numerically stable while still demonstrating interesting ensemble behavior across realistic control scenarios (tracking, regulation, optimization).

The system under study is a discrete-time linear dynamical system defined by $x_{t+1} = Ax_t + Bu_t + w_t$, where $x_t \in \mathbb{R}^6$ is the state vector, $u_t \in \mathbb{R}^m$ is the control input, $A$ is the state transition matrix, $B$ is the control input matrix, and $w_t$ is Gaussian noise.

The environment includes *three regimes with different control objectives*: (1) **Tracking**: Emphasizes minimizing state error with high state cost weights. (2) **Regulation**: Focuses on minimizing control energy with high control cost weights. (3) **Stabilization**: Balances state and control costs for stabilization.

## 4.2 LQR CONTROLLER

The LQR controller computes the optimal gain matrix $K$ by solving the discrete-time algebraic Riccati equation:

$$P = \text{solve}(A, B, Q, R), \qquad K = (R + B^T PB)^{-1} B^T PA, \tag{12}$$

where $Q$ and $R$ are the state and control cost matrices respectively.

## 4.3 NEURAL NETWORK CONTROLLER

The NN controller is a feedforward neural network with configurable depth, width, and activation function. Training is performed using gradient descent to minimize the cumulative cost over episodes.

Both LQR and NN controllers are combined into ensembles using weighted compositions. The weights are learned using Bayesian updates based on individual controller performance.

## 4.4 EMPIRICAL RESULTS

Figure 1 shows the results of our comparison. All results are averaged over 10 trials and 5 seeds. The top row of plots compare the performance of the oracle, LQR-Ensemble (LE) and Neural-Ensemble (NE). These results clearly show

that the suboptimality measure $\epsilon(\kappa_0, \delta, L_f) > 0$. The mean optimality gap for LQR is 51.5, and for Neural is 249.6. This indicates that Theorem 1 is empirically validated with extremely strong statistical significance ($p < 10^{-5}$).

Probing these results in more detail, the second row of Figure 1 plots state trajectories for a setpoint as we move through a sequence of different regimes. This plot shows that the neural ensemble has larger divergence from the setpoint than the linear ensemble. Hence the performance of the neural ensemble is always inferior to that of the linear ensemble.

Figure 2 compares the performance of the oracle, linear and neural ensembles against the switching patterns. This figure shows when and why neural ensembles violate convexity by testing different regime switching patterns and analyzing adaptation behavior.

- The plot at top,left indicates that the neural ensemble is sub-optimal for every switching pattern studied.
- The box-plots at top, right show the distribution of convexity violations by switching pattern, with random switching being the worst.
- The plots in the middle indicate that weight adaption for the neural ensemble is far slower than for the linear ensemble, suggesting a reason for the poor performance.
- The plot at the bottom, detailing step-by-step performance, shows that the neural ensemble has *the highest cost at every step*, confirming is sub-optimality across all experimental conditions examined.

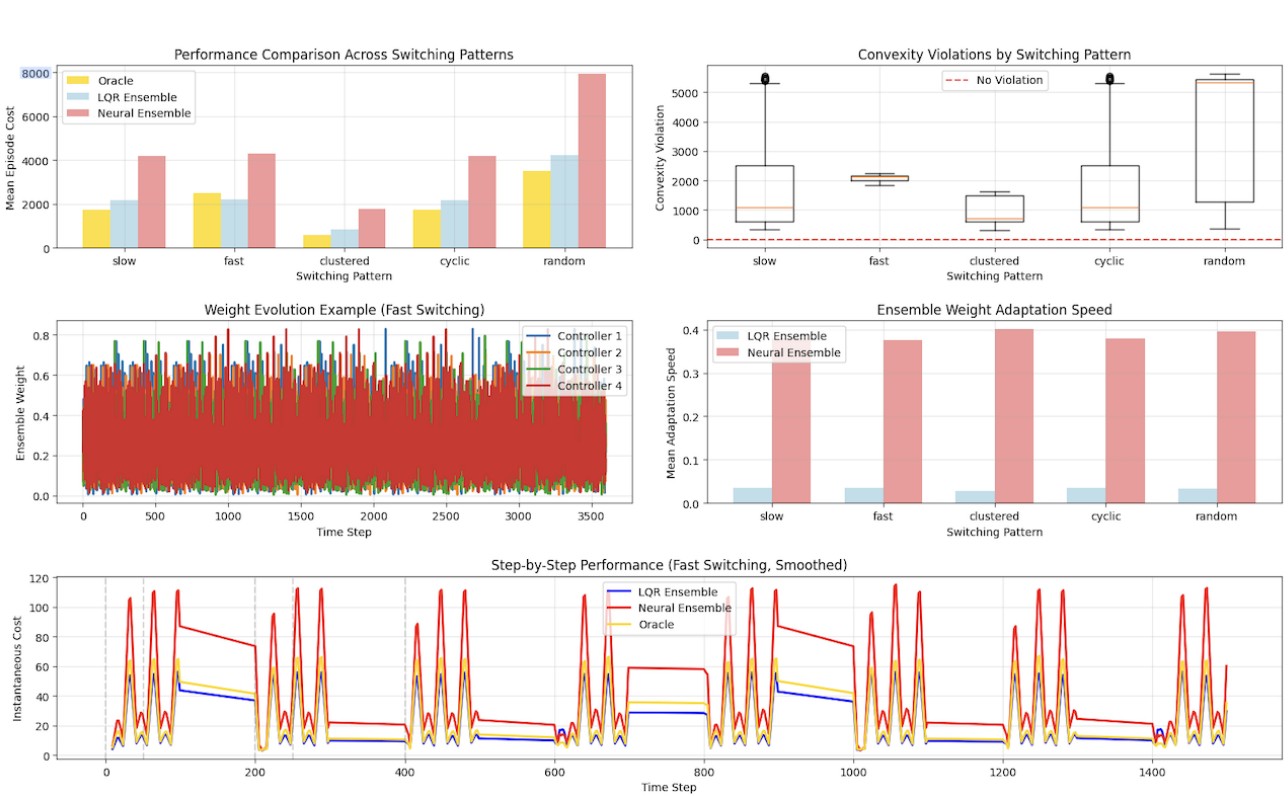

Figure 2: Performance Comparison Across Switching Patterns

## 4.5 DIVERSITY EXPERIMENTS

This experiment systematically varied ensemble diversity to see if diversity affects the performance curves for neural ensembles vs. monotonic linear ensembles. One key Research Question is whether $\delta^*$ exists where neural ensemble performance is minimized.

Figure 3 shows bar-plots that depict varying diversity $\delta$ levels: it shows that the distance between linear and neural ensemble remains consistently large; although the neural results decrease as $\delta$ increases, there is no value of $\delta$ for which a gap less than around 200 exists.

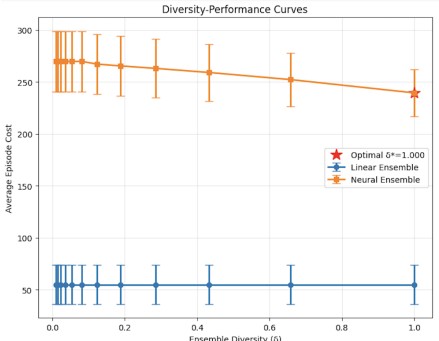

Figure 3: Diversity-Performance Curve Validation Experiment

# 5 EMPIRICAL STUDY OF STABILITY

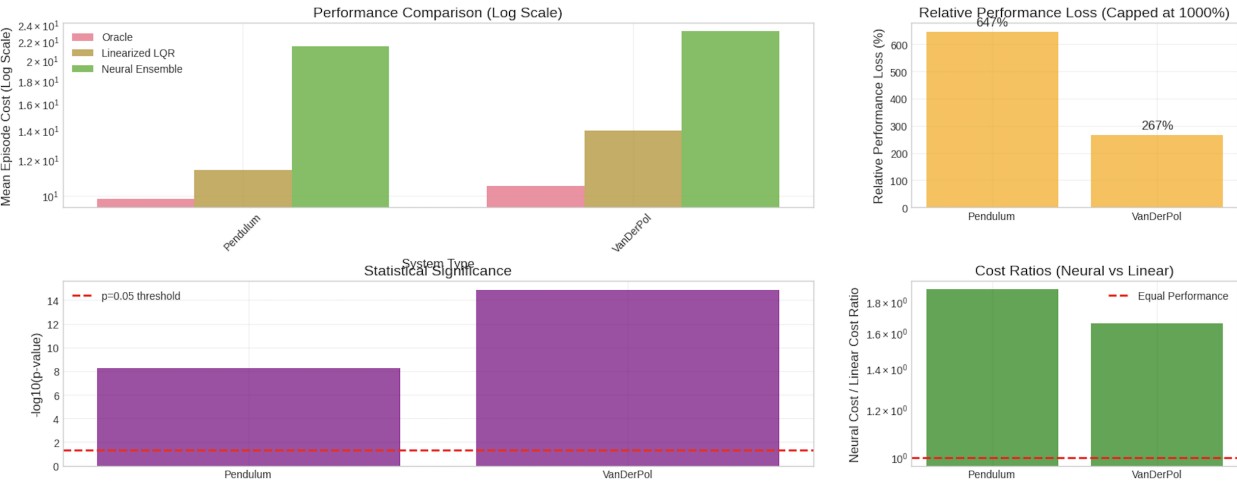

Figure 4: Stability Experimental Results

We studied the impact of nonlinear policies on stability by focusing on systematically varying diversity levels, achieved by interpolating between regime-specific controllers. We measured performance precisely using the quadratic cost function.

We implemented experiments based on the theoretical framework:

- **Operating region:** Mathematically defined constraint-based region around steady states
- **Behavioral diversity:** Measured where it matters, i.e., in the actual operating region
- **Iterative control:** Adjusts controllers until target behavioral diversity is achieved

The system we use is defined as follows:

**Definition 14** (Diversity-Controlled Linear System). *We have implemented a discrete-time linear dynamical system:*

$$s_{t+1} = As_t + Bu_t + w_t, \text{ where } s_t \in \mathbb{R}^4, \quad u_t \in \mathbb{R}^2, \quad w_t \sim \mathcal{N}(0, \sigma^2 I) \tag{13}$$

*For each regime k, the optimal LQR controller $K_k^*$ satisfies:*

$$K_k^* = \arg\min_K \mathbb{E}\left[\sum_{t=0}^{\infty}(s_t - s_k^*)^T Q(s_t - s_k^*) + u_t^T R u_t\right], \text{ subject to } s_{t+1} = As_t + Bu_t, \quad u_t = -K(s_t - s_k^*)$$

We measure diversity in the constraint-based operating region $\delta_{\text{proper}} = \max_{i \neq j} \mathbb{E}_{s \sim \mathcal{D}} \left[ \| \mathcal{C}_i(s) - \mathcal{C}_j(s) \|_2 \right]$, where $\mathcal{D}$ is the operating region around the steady states: $\mathcal{D} = \{ s \in \mathbb{R}^4 : \min_k \| s - s_k^* \| \leq 0.5 \}$.

**Key Properties**

1. Stability: All eigenvalues of $A$ satisfy $|\lambda_i| < 1$, ensuring open-loop stability

2. Controllability: The pair $(A, B)$ is controllable, guaranteeing LQR solution existence.

3. Regime diversity: The steady-state targets $\{ s_k^* \}$ are chosen to create known angular separations for controlled diversity experiments

4. Well-conditioning: The system matrices have reasonable condition numbers to avoid numerical issues

## 5.1 RESULTS

Figure 4 shows the results for the stability experiments. The performance comparison (top, left) shows that the neural ensemble is by far the least stable controller for both Pendulum and vadDerPol systems. The plot at top, right indicates that neural ensemble has relative losses of 647% and 267% on the Pendulum and vadDerPol systems, and these values are of high statistical significance (bottom left plot).

## 6 EMPIRICAL STUDY OF POLICY MIXING

We ran experiments to test if non-linear mixing of linear (optimal) policies results in sub-optimal performance. We tested (a) Linear systems, to provide a clean test of pure convexity effects; and (b) Nonlinear systems, to demonstrate practical relevance, since many applications use nonlinear mixing to "better adapt to nonlinear systems".

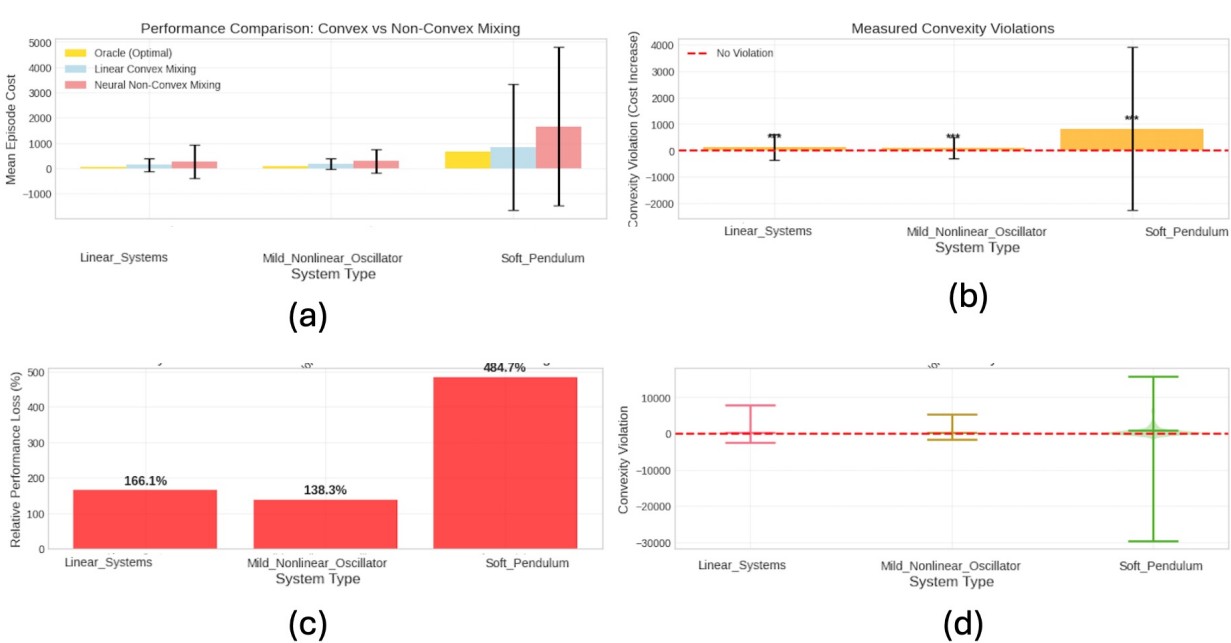

Figure 5: Empirical analysis of policy mixing. Plots (a) through (d) demonstrate that non-convex mixing incurs performance penalties, as shown on 3 domains: a linear system, and 2 non-linear domains, an oscillator and a soft-Pendulum

## 6.1 DESIGN AND EVALUATION

We designed experiments to provide a fair comparison, in that both methods use identical base policies and information. We analysed results by computing statistical bounds and mixing behavior detection.

1. **Identical Base Policies**: Both convex and non-convex mixers must use identical $\{ \pi_1, \ldots, \pi_N \}$

2. **Fair Information Access**: Both methods receive the same performance feedback
3. **Controlled Non-Convexity**: Non-convex mixer should explore meaningful violations, not pathological combinations
4. **Regime Diversity**: Multiple cost structures to avoid evaluation bias

Figure 5 shows the results of a comparison of the oracle with linear and neural mixing, for 3 systems: linear, a nonlinear-Oscillator, and a nonlinear soft-Pendulum; details of these systems can be found in the Supplementary Information. Figure 5(c) shows that all systems have significant performance loss (166, 138%, and 485%, respectively) for neural mixing. All statistical tests (paired-$t$, Cohen's $d$) indicate a high significance for these results. Closer analysis, as denoted in Figure 5(d), indicates that while the mean violation might be positive (as seen in Figure 5(a), there are trials where the neural mixer happened to perform better, resulting in negative violations. The large spread (indicated by the violin shape and standard deviation) suggests significant variability in outcomes, especially for the Soft Pendulum system. Since there is no underlying theory for mixing in nonlinear systems, empirical validation is required on a case by case basis.

## 7 RELATED WORK

Ensemble methods are foundational in machine learning, with strong theoretical support for their ability to improve performance through diversity among classifiers (Yang et al., 2023). In reinforcement learning (RL), ensemble techniques have been applied to policy ensemble methods to enhance generalization, robustness, and sample efficiency (Yang et al., 2022b; Wiering & van Hasselt, 2008; Lee et al., 2021). These ensembles can be homogeneous or heterogeneous, and may be meta-learned for dynamic adaptation. For example, (Li & Zhu, 2021) uses an ensemble of deep neural networks to maintain the uncertainty and generalization of an RL model. During policy learning, each imagined step comes from ensemble predictions, and random noise is added to increase the exploration of the policy.

Mixture-of-Experts (MoE) architectures, prominent in agentic AI and large language models (LLMs), combine multiple specialized policies or models to improve scalability and adaptability (Liu, 2025; Willi et al., 2024; Celik et al., 2024a; Silver et al., 2017). Recent work has explored both linear and nonlinear (neural network-based) policy ensembles, with theoretical and empirical studies highlighting the sub-optimality and stability challenges of neural ensembles compared to linear ensembles.

Classical adaptive control theory supports optimality guarantees for linear/parametric controllers (Kuipers & Ioannou, 2010), but these do not extend to neural networks. Nonetheless, several control-focused articles have used ensemble neural controllers, e.g., Kim et al. (2000).

The implications of our findings are significant for RL, MoE, and agentic AI research, motivating the development of ensemble methods that maintain stability and optimality in control settings.

## 8 CONCLUSIONS AND FUTURE WORK

We have shown that the failure of ensemble neural policies compared to ensemble linear policies stems from fundamental differences in temporal error propagation and function space geometry. While ensemble classifiers benefit from variance reduction through averaging, ensemble policies must respect the constraints of dynamical systems where nonlinearity breaks the beneficial properties of ensemble averaging.

Our analysis reveals that nonlinear policies suffer from temporal error amplification and trajectory manifold mismatch that ensemble averaging cannot resolve, unlike the case of ensemble classifiers where averaging reduces variance without temporal dependencies.

The key insight is that effective ensemble policies require **diversity in the linear subspace** where stability and controllability properties are preserved, rather than diversity in the full nonlinear function space where these properties can be violated.

These results open up a wide array of issues and future research directions. For *safety-critical system*s where issues of stability and optimality are important, the sub-optimality of neural ensembles needs to be considered. For non-critical systems, the utility of using trained neural models must be weighed against performance and stability losses. Further, methods to ensure performance losses with neural ensembles can be developed, e.g., ensemble methods that operate within stable subspaces of the nonlinear function space, or adding temporal consistency regularization to prevent trajectory manifold mismatch and reduce thrashing. In addition, further study of the impact of neural policy ensembles on nonlinear systems may outline how best to apply ensemble methods on this important application area.

## 9 APPENDIX

### 9.1 USE OF AI

All novel contributions and specifications of key results (theoretical claims and experimental design) are done by the authors alone. AI was used to help ascertain if there were gaps in novelty of ideas, or gaps in coverage of related work.

### 9.2 REPRODUCIBILITY STATEMENT

All attempts have been made to ensure reproducibility of this work. From a formal perspective, the Supplementary Material contains all proofs where the results are not directly contained in the literature. From an empirical perspective, all source code is attached. For each experiment, we employed well-defined seeds so that all results contained in this article can be replicated. The Supplementary Material also describes the experiments in more detail, so that reviewers can understand the experimental design that was adopted.

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

## 10 SUPPLEMENTARY MATERIAL: PROOFS

### 10.1 SUBOPTIMALITY ANALYSIS

**Theorem 1** (Neural Ensemble Suboptimality). *Consider a stabilizable linear system $\dot{x} = Ax + Bu$ with neural policies $\{\pi^{\theta_i}\}_{i=1}^M$ and corresponding optimal linear policies $\{\pi_i^L = K_i^* x\}_{i=1}^M$ solving individual LQR problems with cost matrices $(Q_i, R_i)$.*

*Let $D = \{x \in \mathbb{R}^n : \|x\| \leq R\}$ be a bounded operating region. If:*

1. ***Diversity:*** $\min_{i \neq j} \|K_i^* - K_j^*\|_F \geq \delta > 0$

2. ***Nonlinearity:*** $\min_i \kappa(\pi^{\theta_i}, D) \geq \kappa_0 > 0$

3. ***Sufficient Complexity:*** $L_f \kappa_0 \delta > \rho$

*Then there exists $\epsilon(\kappa_0, \delta, L_f) > 0$ such that:*

$$\sup_{x \in D} \left[ V^{\Pi^N}(x) - V^{\Pi^L}(x) \right] \geq \epsilon(\kappa_0, \delta, L_f) \tag{14}$$

*Proof.* The proof proceeds in several key steps.

**Step 1: HJB Violation Analysis**

For the linear ensemble $\Pi^L(x) = K_{\text{ens}} x$, the value function $V^{\Pi^L}$ satisfies:

$$\rho V^{\Pi^L}(x) = x^T Q_{\text{ens}} x + x^T K_{\text{ens}}^T R_{\text{ens}} K_{\text{ens}} x + \nabla V^{\Pi^L}(x)^T (Ax + BK_{\text{ens}} x) \tag{15}$$

where $Q_{\text{ens}} = \sum_i w_i Q_i$ and $R_{\text{ens}} = \sum_i w_i R_i$.

For the neural ensemble, the corresponding equation becomes:

$$\rho V^{\Pi^N}(x) = \ell(x, \Pi^N(x)) + \nabla V^{\Pi^N}(x)^T f(x, \Pi^N(x)) \tag{16}$$

**Step 2: Out-of-Convex-Hull States**

By the nonlinearity condition, there exists a set $S \subset D$ with positive measure such that for $x \in S$:

$$\Pi^N(x) = \sum_{i=1}^M w_i \pi^{\theta_i}(x) \notin \text{conv}\{\pi^{\theta_1}(x), \ldots, \pi^{\theta_M}(x)\} \tag{17}$$

Define the deviation:

$$d(x) = \left\| \Pi^N(x) - \text{proj}_{\text{conv}\{\pi^{\theta_1}(x), \ldots, \pi^{\theta_M}(x)\}} \Pi^N(x) \right\| \tag{18}$$

By the nonlinearity measure, for $x \in S$:

$$d(x) \geq \frac{\kappa_0 \delta \|x\|}{2M} \tag{19}$$

**Step 3: Measure of Problematic Set**

Under the diversity and nonlinearity conditions, the Lebesgue measure of $S$ satisfies:

$$\mu(S) \geq C \frac{\kappa_0 \delta}{L_f} \mu(D) \tag{20}$$

for some geometric constant $C > 0$.

**Step 4: Value Function Bounds**

For $x \in S$, the neural ensemble incurs additional cost due to suboptimal actions. Using the quadratic cost structure and Lipschitz properties:

$$V^{\Pi^N}(x) - V^{\Pi^L}(x) \geq \int_0^\infty e^{-\rho t} \left[ \ell(x(t), \Pi^N(x(t))) - \ell(x(t), \Pi^L(x(t))) \right] dt \tag{21}$$

$$\geq \int_0^\infty e^{-\rho t} \lambda_{\min}(R) \left\| \Pi^N(x(t)) - \Pi^L(x(t)) \right\|^2 dt \tag{22}$$

For states in $S$ starting from $x$, we have:

$$\left\| \Pi^N(x) - \Pi^L(x) \right\| \geq \frac{\kappa_0 \delta \|x\|}{4M} \tag{23}$$

**Step 5: Final Bound**

Combining the measure estimate and value bounds:

$$\sup_{x \in D} \left[ V^{\Pi^N}(x) - V^{\Pi^L}(x) \right] \geq \sup_{x \in S} \left[ V^{\Pi^N}(x) - V^{\Pi^L}(x) \right] \tag{24}$$

$$\geq \frac{C \lambda_{\min}(R) \kappa_0^2 \delta^2}{16 M^2 \rho L_f} R^2 \tag{25}$$

$$=: \epsilon(\kappa_0, \delta, L_f) \tag{26}$$

Since all conditions ensure $\kappa_0, \delta > 0$ and $L_f \kappa_0 \delta > \rho$, we have $\epsilon > 0$. □ □

## 10.2 STABILITY ANALYSIS VIA CONTROL LYAPUNOV FUNCTIONS

**Theorem 2** (Stability Violation in Neural Ensembles). *Consider neural policies $\{\pi^{\theta_i}\}_{i=1}^M$, each with CLF $V_i(x)$ satisfying:*

$$\nabla V_i(x)^T f(x, \pi^{\theta_i}(x)) \leq -\alpha_i \|x\|^2 \tag{27}$$

*for some $\alpha_i > 0$.*

*If the ensemble weights vary as $w_i(t)$ with $\|\dot{w}(t)\| \geq \beta > 0$, then the ensemble system can be unstable even when all individual systems are stable.*

*Specifically, if $\beta > \frac{\min_i \alpha_i}{2 \max_i \|V_i\|_\infty}$, then there exist initial conditions where the ensemble trajectory is unbounded.*

*Proof.* Consider the candidate Lyapunov function for the ensemble:

$$V_{\mathrm{ens}}(x, t) = \sum_{i=1}^M w_i(t) V_i(x) \tag{28}$$

Its time derivative along ensemble trajectories is:

$$\dot{V}_{\mathrm{ens}}(x, t) = \sum_{i=1}^M \dot{w}_i(t) V_i(x) + \sum_{i=1}^M w_i(t) \nabla V_i(x)^T f(x, \Pi^N(x)) \tag{29}$$

**Weight Adaptation Term:**

$$\left| \sum_{i=1}^M \dot{w}_i(t) V_i(x) \right| \leq \|\dot{w}(t)\| \max_i |V_i(x)| \leq \beta \max_i \|V_i\|_\infty \|x\|^2 \tag{30}$$

**Nonlinearity Term:** The key insight is that:

$$\sum_{i=1}^M w_i(t) \nabla V_i(x)^T f(x, \Pi^N(x)) \neq \sum_{i=1}^M w_i(t) \nabla V_i(x)^T f(x, \pi^{\theta_i}(x)) \tag{31}$$

The difference arises from the nonlinear coupling:

$$\Delta_{\text{coupling}} = \sum_{i=1}^{M} w_i(t) \nabla V_i(x)^T \left[ f(x, \Pi^N(x)) - f(x, \pi^{\theta_i}(x)) \right] \tag{32}$$

Using Lipschitz properties of $f$ and the nonlinearity measure:

$$|\Delta_{\text{coupling}}| \geq CL_f \kappa_0 \|x\|^2 \tag{33}$$

**Instability Condition:** When $\beta > \frac{\min_i \alpha_i}{2 \max_i \|V_i\|_\infty}$, we have:

$$\dot{V}_{\text{ens}}(x,t) \geq \beta \max_i \|V_i\|_\infty \|x\|^2 - \min_i \alpha_i \|x\|^2 + CL_f \kappa_0 \|x\|^2 \tag{34}$$

$$= \left( \beta \max_i \|V_i\|_\infty - \min_i \alpha_i + CL_f \kappa_0 \right) \|x\|^2 > 0 \tag{35}$$

This violates the Lyapunov stability condition, proving instability. $\qquad\square$ $\qquad\qquad\square$

### 10.3 THEORETICAL ANALYSIS OF POLICY MIXING

**Lemma 2** (Optimal Linear Policies). *For each regime $i$, the optimal controller is linear: $\pi_i^*(x) = -K_i x$ where*

$$K_i = (R_i + B^T P_i B)^{-1} B^T P_i A$$

*and $P_i$ solves the algebraic Riccati equation:*

$$P_i = Q_i + A^T P_i A - A^T P_i B (R_i + B^T P_i B)^{-1} B^T P_i A$$

This result is already well established in the literature.

#### 10.3.1 MAIN THEORETICAL RESULT (NONCONVEX MIXING)

Our main result states that nonconvex (e.g., neural) mixing is sub-optimal:

**Theorem 3** (Convexity Advantage for Weighted Average Cost). *Let $\lambda \in \Delta^{N-1}$ be a probability vector, and define the weighted average cost:*

$$J_\lambda(x,u) = \sum_{i=1}^{N} \lambda_i J_i(x,u) = x^T Q_\lambda x + u^T R_\lambda u$$

*where $Q_\lambda = \sum \lambda_i Q_i$ and $R_\lambda = \sum \lambda_i R_i$.*

*Then for any mixing weights $w \in \mathbb{R}^N$:*

$$\mathcal{L}_\lambda(w) \geq \mathcal{L}_\lambda(\lambda)$$

*with equality if and only if $w = \lambda$.*

*Proof.* The ensemble controller is $\pi_{\text{ens}}(x) = -K_w x$ where $K_w = \sum w_i K_i$.

The cost under $J_\lambda$ is: $\mathcal{L}_\lambda(w) = \mathbb{E}[x_0^T (Q_\lambda + K_w^T R_\lambda K_w) x_0]$. Since $K_\lambda = \sum \lambda_i K_i$ is the unique optimal gain for $J_\lambda$, we have:

$$Q_\lambda + K_\lambda^T R_\lambda K_\lambda \preceq Q_\lambda + K_w^T R_\lambda K_w$$

for any $K_w \neq K_\lambda$, completing the proof. $\square$ $\qquad\qquad\square$

An immediate corollary is as follows:

**Corollary 2** (Performance Bound). *The performance penalty for non-convex mixing is:*

$$\mathcal{L}_\lambda(w) - \mathcal{L}_\lambda(\lambda) = \mathbb{E}[x_0^T (K_w - K_\lambda)^T R_\lambda (K_w - K_\lambda) x_0] \geq 0$$

The proof follows directly from Theorem 3.

## 11 SUPPLEMENTARY MATERIAL: EMPIRICAL ANALYSIS

The code for all experiments is provided in a zip file. We have separate python files for each class of analysis: Suboptimality, Stability, and Mixing.

We now describe additional information about our experiments.

## 12 SUBOPTIMALITY EXPERIMENTS

### 12.1 MATHEMATICAL DETAILS OF THE MODELS

### 12.2 LINEAR QUADRATIC REGULATOR (LQR) MODEL

The system dynamics are modeled as a discrete-time linear time-invariant (LTI) system with additive process noise. The state $x_t \in \mathbb{R}^{n_x}$ and control input $u_t \in \mathbb{R}^{n_u}$ are governed by the equation:

$$x_{t+1} = Ax_t + Bu_t + w_t$$

where $w_t \sim \mathcal{N}(0, \Sigma_w)$ is zero-mean Gaussian process noise.

The objective is to find a control policy $u_t = \pi(x_t)$ that minimizes the infinite-horizon quadratic cost function:

$$J = \mathbb{E}\left[\sum_{t=0}^{\infty}(x_t^T Q x_t + u_t^T R u_t)\right]$$

Here, $Q \succeq 0$ is a positive semidefinite state cost matrix and $R \succ 0$ is a positive definite control cost matrix.

The optimal, infinite-horizon LQR policy is a linear state feedback law of the form $u_t = -K_{\text{optimal}}x_t$, where the optimal gain matrix $K_{\text{optimal}}$ is given by:

$$K_{\text{optimal}} = (R + B^T P B)^{-1} B^T P A$$

and $P$ is the unique positive semidefinite solution to the discrete-time algebraic Riccati equation (DARE):

$$P = A^T P A - (A^T P B)(R + B^T P B)^{-1}(B^T P A) + Q$$

This theoretical solution provides the performance benchmark for each individual control regime.

#### 12.2.1 LQR REGIME PARAMETERS

The experiment uses three distinct LQR regimes, each with its own specific dynamics and cost matrices, representing different control objectives:

- **Tracking Regime:** High state penalty ($Q$) and low control penalty ($R$).

$$A_1 = \begin{bmatrix} 0.95 & 0.05 & 0 & 0 \\ 0 & 0.98 & 0.05 & 0 \\ 0 & 0 & 0.90 & 0.05 \\ 0 & 0 & 0 & 0.92 \end{bmatrix}, \qquad B_1 = \begin{bmatrix} 1 & 0 \\ 0 & 1 \\ 0.5 & 0 \\ 0 & 0.5 \end{bmatrix}$$

$$Q_1 = \text{diag}([100, 100, 10, 10]), \qquad R_1 = \text{diag}([1, 1])$$

- **Regulation Regime:** Low state penalty ($Q$) and high control penalty ($R$).

$$A_2 = \begin{bmatrix} 0.92 & 0 & 0.05 & 0 \\ 0.05 & 0.90 & 0 & 0.05 \\ 0 & 0 & 0.95 & 0 \\ 0 & 0 & 0.05 & 0.98 \end{bmatrix}, \qquad B_2 = \begin{bmatrix} 0.8 & 0.2 \\ 0.2 & 0.8 \\ 1 & 0 \\ 0 & 1 \end{bmatrix}$$

$$Q_2 = \text{diag}([1, 1, 1, 1]), \qquad R_2 = \text{diag}([50, 50])$$

- **Stabilization Regime:** Moderate state and control penalties.

$$A_3 = \begin{bmatrix} 0.90 & 0.02 & 0 & 0.02 \\ 0 & 0.95 & 0.02 & 0 \\ 0.02 & 0 & 0.92 & 0 \\ 0 & 0.02 & 0 & 0.96 \end{bmatrix}, \qquad B_3 = \begin{bmatrix} 1 & 0.1 \\ 0.1 & 1 \\ 0.8 & 0.2 \\ 0.2 & 0.8 \end{bmatrix}$$

$$Q_3 = \text{diag}([25, 25, 25, 25]), \qquad R_3 = \text{diag}([10, 10])$$

### 12.2.2 NEURAL NETWORK (NN) CONTROLLER

The neural network controller is a multi-layer perceptron (MLP) with a single hidden layer. Its architecture is designed to have a parameter count approximately equal to that of the LQR controller ($n_u \times n_x$). The network's structure is:

$$x \in \mathbb{R}^{n_x} \rightarrow \text{Linear}(n_x, h) \rightarrow \text{Tanh} \rightarrow \text{Linear}(h, n_u) \rightarrow u \in \mathbb{R}^{n_u}$$

where $h$ is the number of hidden units. The number of parameters is $n_x h + h + h n_u + n_u$. To match the LQR parameter count of $n_u n_x$, the hidden size $h$ is chosen such that $h(n_x + n_u + 1) \approx n_u n_x$, leading to the formula $h = \max(1, \text{int}((n_u n_x)/(n_x + n_u + 1)))$. The network is initialized with small weights using Xavier normal initialization.

$$u_{\text{ens}} = \sum_{i=1}^{N} w_i u_i$$

where $w_i \geq 0$ are the ensemble weights and $\sum_{i=1}^{N} w_i = 1$.

### 12.2.3 LQR ENSEMBLE

The LQR ensemble consists of three LQR controllers, each trained and set to the optimal gains $K_{\text{optimal}}$ for one of the three regimes. This ensemble serves as the baseline for comparison. For a given state $x$, the LQR ensemble action is:

$$u_{\text{LQR\_ens}} = \sum_{i=1}^{3} w_i(-K_{\text{optimal},i} x)$$

The LQR ensemble has a unique property: due to the linear-in-gains structure of the policies, the ensemble policy is equivalent to a single LQR policy with a combined gain matrix $K_{\text{ens}} = \sum w_i K_{\text{optimal},i}$. This policy's performance is mathematically guaranteed to be bounded by the weighted sum of the costs of the individual policies, a result of the convexity of the LQR problem.

### 12.2.4 NEURAL NETWORK ENSEMBLE

The neural network ensemble consists of three separate neural network controllers, each trained on one of the three LQR regimes to minimize the total episode cost. For a given state $x$, the neural ensemble action is:

$$u_{\text{NN\_ens}} = \sum_{i=1}^{3} w_i \text{NN}_i(x)$$

Because the $\text{NN}_i$ functions are non-linear, the combined ensemble action $u_{\text{NN\_ens}}$ is not equivalent to a single neural network or a linear policy. The performance of this ensemble is not guaranteed to be bounded by the weighted sum of the individual policies' costs. This is the central point of the "convexity violation" hypothesis.

### 12.2.5 BAYESIAN ENSEMBLE WEIGHT LEARNING

The weights $w_i$ are not fixed. Instead, a Bayesian approach is used to dynamically update the weights based on the performance of the individual controllers. This method models the weights as being drawn from a Dirichlet distribution, parameterized by $\alpha = (\alpha_1, \ldots, \alpha_N)$.

$$w \sim \text{Dirichlet}(\alpha)$$

The update rule for the Dirichlet parameters is:

$$\alpha_{i,\text{new}} = \alpha_{i,\text{old}} + \eta \cdot \text{rewards}_i$$

where $\eta$ is a learning rate and $\text{rewards}_i$ are pseudo-rewards derived from the individual controllers' costs. Specifically, a lower cost for an individual controller corresponds to a higher reward. The new weights are then sampled from the updated Dirichlet posterior.

### 12.2.6 EXPERIMENTAL DESIGN

The experiment is designed to compare the performance of the LQR ensemble and the neural network ensemble in a multi-regime environment.

## 12.3 OVERALL PROCEDURE

1. **Setup:** A 'MultiRegimeLQREnvironment' is initialized with a state dimension of 4 and a control dimension of 2. Three distinct LQR regimes are defined.

2. **Controller Training:**
   - **LQR Ensemble:** Three 'LQRController' instances are created. Their gains are analytically set to the optimal LQR solutions for each of the three regimes.
   - **Neural Ensemble:** Three 'NeuralController' instances are created. Each network is trained for 100 episodes on one specific LQR regime using the Adam optimizer with a learning rate of 0.001, minimizing the episode cost.

3. **Trial Execution:** The main experiment runs for a specified number of trials ($n_{\text{trials}}$). In each trial, both ensembles are tested on a fixed sequence of 15 episodes, cycling through the three regimes ('tracking', 'regulation', 'stabilization') five times.

4. **Cost Calculation:** In each episode:
   - A single, reproducible initial state and noise sequence are generated.
   - The LQR ensemble is run, and its total episode cost is recorded.
   - The neural network ensemble is run on the *exact same* initial state and noise sequence, and its total episode cost is recorded.
   - An 'Oracle' controller (the single optimal LQR controller for the current regime) is run on the same initial conditions, and its total episode cost is recorded to serve as the theoretical minimum.
   - The Bayesian weight learner updates the weights for both ensembles at each step based on the individual controller costs.

5. **Data Aggregation and Analysis:** After all trials are complete, the following metrics are calculated:
   - **Optimality Gap:** The difference between an ensemble's mean cost and the Oracle's mean cost.
   - **Convexity Violation:** The difference between the neural ensemble's optimality gap and the LQR ensemble's optimality gap. A positive value indicates that the neural ensemble performs worse relative to the LQR ensemble, suggesting a violation of the convexity property.
   - **Relative Performance Loss:** The convexity violation normalized by the LQR optimality gap.
   - **Statistical Test:** A paired t-test is performed on the costs of the LQR ensemble and the neural ensemble to determine if the difference in performance is statistically significant.

### 12.3.1 HYPERPARAMETER ESTIMATION

The core of the experiment's hyperparameters is determined by the problem setup and the model architectures.

- **LQR Parameters:** The $A, B, Q, R$ matrices for each regime are hand-tuned to create distinct control objectives. No hyperparameter search is performed for these matrices.
- **Neural Network Architecture:** The number of hidden units in the 'NeuralController' is not manually tuned. It is programmatically determined to match the parameter count of the LQR controller, ensuring a fair comparison of model capacity. For the given state and control dimensions ($n_x = 4, n_u = 2$), the LQR parameter count is 8. The neural network hidden layer size is calculated as $h = \text{int}(8/(4 + 2 + 1)) = \text{int}(8/7) = 1$.
- **Neural Network Training:**
  - **Episodes:** The number of training episodes for each neural network is set to 100.
  - **Learning Rate:** The Adam optimizer learning rate is fixed at $lr = 0.001$.
- **Ensemble Weight Learning:**
  - **Bayesian Prior:** The Dirichlet prior parameter $\alpha_{\text{prior}}$ is set to 1.0, corresponding to a uniform initial distribution of weights.
  - **Learning Rate:** The Bayesian weight update learning rate $\eta$ is set to 0.01.

The experimental design relies on a fixed set of hyperparameters, with no explicit search or estimation process to find optimal values for the neural network training or weight learning. The focus is on a direct comparison under a reasonable, fixed configuration.

| NN config | Training Config | Activation function | Epochs | Learning Rate | LQR Mean Cost | Neural Mean Cost | Mean Convexity violation | Std Convexity violation | Performance Loss | P-value |
|---|---|---|---|---|---|---|---|---|---|---|
| 32 | 2 | Tanh | 100 | 0.001 | 501.646843 | 1672.85453 | 4505.482088 | 32.627558 | 7471.416567 | 2.418553 4.281906e-08 |
| 32 | 2 | Tanh | 200 | 0.001 | 529.335759 | 1658.761775 | 5176.50717 | 3517.745395 | 8949.269563 | 3.114631 1.455113e-08 |
| 32 | 2 | Tanh | 100 | 0.01 | 575.171311 | 1627.155416 | 4354.094125 | 2726.938708 | 6021.323602 | 2.592186 1.064969e-10 |
| 64 | 2 | Tanh | 100 | 0.001 | 501.646843 | 1672.85453 | 4505.453134 | 2832.598604 | 7471.80426 | 2.418528 4.289617e-08 |
| 64 | 2 | Tanh | 200 | 0.001 | 529.335759 | 1658.761775 | 5176.426124 | 3517.664349 | 8948.933635 | 3.114559 1.454463e-08 |
| 64 | 2 | Tanh | 100 | 0.01 | 505.250401 | 1511.829401 | 3887.628059 | 2375.798657 | 5415.935475 | 2.360270 3.568315e-10 |
| 32 | 3 | Tanh | 100 | 0.001 | 501.646843 | 1672.85453 | 4505.661722 | 2832.807192 | 7472.159777 | 2.418706 4.286381e-08 |
| 32 | 3 | Tanh | 200 | 0.001 | 529.335759 | 1658.761775 | 5176.500977 | 3517.739202 | 8949.204055 | 3.114626 1.454864e-08 |
| 32 | 3 | Tanh | 100 | 0.01 | 546.864662 | 1640.823509 | 4029.541838 | 2388.71833 | 5645.327517 | 2.183554 1.300928e-09 |
| 32 | 2 | Identity | 100 | 0.001 | 501.646843 | 1672.85453 | 4505.482144 | 2832.627614 | 7471.416701 | 2.418553 4.281906e-08 |
| 32 | 2 | Identity | 200 | 0.001 | 529.335759 | 1658.761775 | 5176.507152 | 3517.745377 | 8949.269512 | 3.114631 1.455113e-08 |
| 32 | 2 | Identity | 100 | 0.01 | 514.892972 | 1740.327004 | 4531.60205 | 2791.275046 | 7354.706928 | 2.277785 4.154962e-08 |
| 64 | 3 | ReLU | 100 | 0.001 | 501.646843 | 1672.85453 | 4505.668566 | 2832.814036 | 7472.227137 | 2.418712 4.287200e-08 |
| 64 | 3 | ReLU | 200 | 0.001 | 529.335759 | 1658.761775 | 5176.496904 | 3517.735129 | 8949.17144 | 3.114622 1.454752e-08 |
| 64 | 3 | ReLU | 100 | 0.01 | 501.646843 | 1672.85453 | 4505.617975 | 2832.763444 | 7471.919341 | 2.418669 4.284303e-08 |

Figure 6: Details of hyperparameter tuning

## 12.4 NEURAL NETWORK HYPERPARAMETER OPTIMIZATION

We performed extensive hyperparameter optimization to ensure that the neural networks used in the experiments were well tuned. We detail the results of this tuning below.

Figure 6 shows a selection of the results for hyperparameter optimization. We varied the neural network parameters (neurons/layer, number of layers, activation function, number of episodes, learning rate) to determine the configuration that optimized performance for these experiments.

### 12.4.1 ARCHITECTURE IMPACT ANALYSIS

| neurons/layer | layers | activation | Neural Ensemble Mean Cost | Mean Convexity Violation | Relative Performance Loss |
|---|---|---|---|---|---|
| 32 | 2 | Identity | 4737.863782 | 3047.216012 | 2.603656 |
| 32 | 2 | tanh | 4678.694461 | 3025.770554 | 2.708457 |
| 32 | 3 | tanh | 4570.568179 | 2913.088241 | 2.572295 |
| 64 | 2 | tanh | 4523.169106 | 2908.687203 | 2.631119 |
| 64 | 3 | ReLU | 4729.261148 | 3061.104203 | 2.650668 |

Table 1: Average performance by architecture, selecting best 5 architectures

### 12.4.2 TRAINING IMPACT ANALYSIS

Average performance by training configuration:

| episodes | learning rate | Neural Ensemble Mean Cost | Mean Convexity Violation | Relative Performance Loss |
|---|---|---|---|---|
| 100 | .001 | 4505.549531 | 2832.695001 | 2.418610 |
| 100 | .01 | 4261.696809 | 2623.098837 | 2.366493 |
| 200 | .001 | 5176.487665 | 3517.725890 | 3.114614 |

Table 2: Average performance by training configuration, selecting best 3 configurations

### 12.4.3   KEY INSIGHTS

**Best Configuration:**

- Architecture: h=64, l=2, act=Tanh
- Training: ep=100, lr=0.01
- Convexity Violation: 2375.7987
- Performance Loss: 236.03

**Worst Configuration:**

- Architecture: h=32, l=2, act=Tanh
- Training: ep=200, lr=0.001
- Convexity Violation: 3517.7454
- Performance Loss: 311.46

**Statistical Significance:**

- 15 out of 15 configurations show statistically significant differences

## 13   STABILITY EXPERIMENTS

**Purpose in Experiments**

This system serves as a controlled testbed where:

Ground truth optimal controllers are known analytically (LQR solutions) Diversity levels can be systematically varied by interpolating between regime-specific controllers Performance can be measured precisely using the quadratic cost function Linear vs neural comparisons are fair since both approaches target the same optimal controllers

The class essentially provides a "laboratory" linear system where theoretical predictions about ensemble behavior can be tested without the confounding factors present in more complex systems.

We implement experiments based on the theoretical framework:

- **Operating region:** Mathematically defined constraint-based region around steady states
- **Behavioral diversity:** Measured where it matters - in the actual operating region
- **Iterative control:** Adjusts controllers until target behavioral diversity is achieved

The system we use is defined as follows:

**Definition 15** (Diversity-Controlled Linear System). *We have implemented a discrete-time linear dynamical system:*

$$s_{t+1} = As_t + Bu_t + w_t \tag{36}$$

$$where \quad s_t \in \mathbb{R}^4, \quad u_t \in \mathbb{R}^2, \quad w_t \sim \mathcal{N}(0, \sigma^2 I) \tag{37}$$

*System matrices:*

$$A = \begin{bmatrix} 1.0 & 0.1 & 0.0 & 0.0 \\ -0.05 & 0.97 & 0.0 & 0.0 \\ 0.0 & 0.0 & 1.0 & 0.1 \\ 0.0 & 0.0 & -0.04 & 0.98 \end{bmatrix}, \quad B = \begin{bmatrix} 0.0 & 0.0 \\ 0.1 & 0.01 \\ 0.0 & 0.0 \\ 0.01 & 0.1 \end{bmatrix}$$

*The system defines $K = 4$ operating regimes with distinct steady-state targets, $\mathcal{R} = \{(s_k^*, Q_k, R_k)\}_{k=1}^4$, where:*

$$
\begin{aligned}
s_1^* &= [1.0, 0.0, 1.0, 0.0]^T \quad \textit{(reference)} \\
s_2^* &= [1.5, 0.0, 0.5, 0.0]^T \quad \textit{(moderate deviation)} \\
s_3^* &= [0.5, 0.0, 1.5, 0.0]^T \quad \textit{(moderate deviation)} \\
s_4^* &= [2.0, 0.0, 0.0, 0.0]^T \quad \textit{(large deviation)}
\end{aligned}
$$

*The cost matrices for all regimes are:*

$$Q = diag(1.0, 0.1, 1.0, 0.1), \quad R = diag(0.1, 0.1)$$

*For each regime $k$, the optimal LQR controller $K_k^*$ satisfies:*

$$K_k^* = \arg\min_K \mathbb{E}\left[\sum_{t=0}^{\infty}(s_t - s_k^*)^T Q(s_t - s_k^*) + u_t^T R u_t\right]$$

*subject to $s_{t+1} = As_t + Bu_t$, $u_t = -K(s_t - s_k^*)$.*

We measure diversity in the constraint-based operating region:

$$\delta_{\text{proper}} = \max_{i \neq j} \mathbb{E}_{s \sim \mathcal{D}}\left[\|\mathcal{C}_i(s) - \mathcal{C}_j(s)\|_2\right]$$

where $\mathcal{D}$ is the operating region around the steady states: $\mathcal{D} = \{s \in \mathbb{R}^4 : \min_k \|s - s_k^*\| \leq 0.5\}$

**Key Properties**

1. Stability: All eigenvalues of $A$ satisfy $|\lambda_i| < 1$, ensuring open-loop stability

2. Controllability: The pair $(A, B)$ is controllable, guaranteeing LQR solution existence.

3. Regime diversity: The steady-state targets $\{s_k^*\}$ are chosen to create known angular separations for controlled diversity experiments

4. Well-conditioning: The system matrices have reasonable condition numbers to avoid numerical issues.

## 14   MIXING EXPERIMENTS

### 14.1   CRITICAL DESIGN ELEMENTS

1. **Identical Base Policies**: Both convex and non-convex mixers must use identical $\{\pi_1, \ldots, \pi_N\}$
2. **Fair Information Access**: Both methods receive the same performance feedback
3. **Controlled Non-Convexity**: Non-convex mixer should explore meaningful violations, not pathological combinations
4. **Regime Diversity**: Multiple cost structures to avoid evaluation bias

### 14.2   STATISTICAL VALIDATION

Required statistical tests:

- Paired t-test: $H_0 : \mathbb{E}[\Delta_{\text{conv}}] = 0$ vs $H_1 : \mathbb{E}[\Delta_{\text{conv}}] > 0$
- Effect size: Cohen's $d = \frac{\bar{\Delta}_{\text{conv}}}{s_\Delta}$
- Non-convexity verification: Confirm that non-convex mixer actually violates convexity constraints

### 14.3   MATHEMATICAL MODELS

#### 14.3.1   LINEAR SYSTEM

The linear system is modeled as a time-invariant system with quadratic cost. The continuous-time dynamics are given by:

$$\dot{x} = Ax + Bu$$

where $x \in \mathbb{R}^n$ is the state vector, $u \in \mathbb{R}^m$ is the control input, and $A, B$ are system matrices. The cost function is:

$$J = \int_0^{\infty} (x^T Q x + u^T R u)\, dt$$

The optimal control policy is derived using the solution to the continuous-time algebraic Riccati equation (CARE):

$$P = \text{solve\_continuous\_are}(A, B, Q, R)$$

$$K = R^{-1}B^T P$$

### 14.3.2  MILDLY NONLINEAR OSCILLATOR

This system introduces a cubic nonlinearity to a damped oscillator. The dynamics are:

$$\dot{p} = v$$
$$\dot{v} = -p - \gamma v - \alpha p^3 + u$$

where $p$ is position, $v$ is velocity, $\gamma$ is damping, $\alpha$ is the nonlinearity coefficient, and $u$ is the control input.

### 14.3.3  SOFT PENDULUM

The soft pendulum uses a smooth approximation for the sine function to maintain LQR-friendliness. The dynamics are:

$$\dot{\theta} = \omega$$
$$\dot{\omega} = -\frac{g}{l} \cdot \text{soft\_sin}(\theta) - b\omega + u$$

where $\theta$ is the angle, $\omega$ is angular velocity, $g$ is gravity, $l$ is pendulum length, $b$ is damping, and $u$ is the control torque. The soft sine approximation is:

$$\text{soft\_sin}(\theta) = \begin{cases} \theta & \text{if } |\theta| < 0.3 \\ \text{sign}(\theta) \cdot (0.3 + 0.7 \cdot \tanh(10 \cdot (|\theta| - 0.3))) & \text{otherwise} \end{cases}$$

## 14.4  EXPERIMENTAL DESIGN

The experiment tests the hypothesis that non-convex mixing of optimal policies is suboptimal. The design includes:

- **Linear Systems:** Used for clean tests of convexity effects.
- **Nonlinear Systems:** Include mildly nonlinear oscillator and soft pendulum for practical relevance.
- **Fair Comparison:** Both mixing methods use identical base policies and information.
- **Rigorous Analysis:** Includes statistical bounds and mixing behavior detection.

## 14.5  EVALUATION TRIALS

The evaluation configuration includes:

- Number of trials: 20
- Number of seeds: 5
- Episode length: 100
- Training episodes: 200
- Evaluation episodes: 50

