# OpenReview forum: "Neural Policy Ensembles are Sub-Optimal"
_ICLR.cc/2026/Conference — Submitted to ICLR 2026_

### Official Review · Reviewer_cH3G · 2025-10-27

**Soundness:** 2
**Presentation:** 1
**Contribution:** 1
**Rating:** 2
**Confidence:** 3

**Summary:**

This paper studies the suboptimality of nonlinear and ensemble policies in continuous control problems. The authors formulate their analysis in the context of linear systems with quadratic costs and compare the performance of nonlinear neural policies and ensembles of neural policies against linear policies and linear ensembles. They define a nonlinearity measure quantifying the deviation of a function from being affine and show that, for linear-quadratic systems, any nonlinear (or non-affine) policy must be suboptimal relative to the optimal linear policy. Theorem 1 and Theorem 2 establish that nonlinear policies cannot be globally optimal, and Theorem 3 extends this argument to ensemble policies, suggesting that non-convex mixtures of policies are also suboptimal compared to convex-mixing policy ensembles. Empirical evaluations are conducted on synthetic linear control systems, comparing the performance of linear, nonlinear, and ensemble policies. The results show that neural policy ensembles perform worse than linear policies and ensembles.

**Strengths:**

- The paper tackles a clear and well-posed theoretical question: under what conditions are neural or nonlinear ensembles suboptimal compared to linear policies? This is question, I believe, is of theoretical interest.
- The mathematical results are formally presented with explicit assumptions and definitions (e.g., the "nonlinearity measure" and the LQR cosntraints). The formalism appears internally consistent and logically valid given the assumptions.
- The paper attempts to connect classical control theory insights (LQR optimality) with modern neural network and ensemble formulations, a potentially valuable contribution.

**Weaknesses:**

**Presentation**:

- I found the paper extremely difficult to parse, both linguistically and conceptually. The writing is often ambiguous, and key terms are not defined well. For example, the opening statement of the abstract ("neural policy ensembles are sub-optimal with respect to linear policy ensembles") is already unclear and leaves the reader guessing what kind of optimality this is referring to.
- The central theoretical claim almost seems self-referential given the assumptions. Since the problem setup is fully linear with quadratic costs, and the class of linear policies already contains the global optimum, it follows by construction that any non-linear function cannot be better. If I understood correctly, this result is highly unsurprising and, arguably, trivial within the stated framework of linear problems with quadratic costs.
- The authors seem to assume that neural policies are nonlinear by construction and therefore suboptimal. However, this neglects the fact that, by the universal approximation theorem, a neural network with ReLU activations can represent a linear function exactly. In other words, a neural policy can approximate the optimal linear controller arbitrarily well. The suboptimality is in my view a statement about nonlinear functions in general, not necessarily about neural policies.
- In the treatment of ensembles, the authors reference, for example, Burke et al. for non-convex mixing with neural networks, but these works consider state-dependent mixtures. In the author's notation, the weights $w_i$ seem to be static rather than state-dependent. As before, the authors exclude the probability simples from the support of non-convex mixing by construction.

**Questions:**

- What motivates the focus on linear dynamics and quadratic costs for the theoretical exposition? It seems clear that LQRs solve these optimally.
- In Theorem 3, how are mixture weights $w_i$ defined as statis scalars or as state-dependent functions? The notation suggests static weights, but the text refers to neural mixing networks.
- How does this result translate to systems that are not linear?

---

> ### Author Response · Authors · 2025-11-17
> **This review contains many significant inaccuracies and misunderstandings**
>
> Thanks to the reviewer for their comments. I would like to  highlight that the review contains many false claims about  the paper and its results. I urge the reviewer to consider the following issues:
>
> To address the question “What motivates the focus on linear dynamics and quadratic costs for the theoretical exposition? It seems clear that LQRs solve these optimally.” I was running experiments on neural policy ensembles and could not get them to work as well as I hoped, and eventually the theory pointed out the results in this paper. This result runs counter to most work on policy ensembles, which use nonlinear functions. So the motivation is clear: the success of classifier ensembles does not mean that policy ensembles work in the same way.
>
> The reviewer claims  that the problem setup is fully linear with quadratic costs, and the class of linear policies already contains the global optimum. This is false. To directly address one question: “How does this result translate to systems that are not linear?” Section 6 presents experimental evidence on nonlinear systems to directly translate to systems that are not linear—-but this is ignored by the reviewer. Additionally, in the main setup of the paper the theory examines an ensemble of nonlinear neural controllers (section 4.3 and throughout the article), as well as the use of nonlinear combination of linear policies.
>
> The claim “the universal approximation theorem, a neural network with ReLU activations can represent a linear function exactly” is technically incorrect. The only activation function that guarantees linearity is a pass-through or linear activation function—ReLU and sigmoid creates a nonlinear function; I am unaware of any literature that uses only linear neural networks, and I ask this reviewer to provide modern references that do NOT use a nonlinear activation function for policy ensembles (or indeed for any real applications). And this is exactly the point raised in the article, i.e., that basically all neural policy models are inherently  nonlinear.
>
> Question 3 states: “In Theorem 3, how are mixture weights  defined as statis scalars or as state-dependent functions?” The mixture weights can be time-dependent, and in fact the classical control method using this mixture approach, MMAC, assumes weights change over time. The suppression of temporal indices for weights is made for simplicity of exposition. All results hold for static or temporal weights.
>
> The claim that “The central theoretical claim almost seems self-referential given the assumptions” is also technically incorrect. The claim “the problem setup is fully linear with quadratic costs” is false—-please indicate where in Definition 12 (Policy Mixing) and Definition 13 (Performance Measure) that linearity requirement is made. There is no linearity requirement in either. The cost function J is arbitrary as is policy $\pi$. The paper explicitly compares arbitrary policy ensembles with linear ensembles.

---

> > ### Comment · Reviewer_cH3G · 2025-11-26
> > **Response to Authors**
> >
> > Thank you for your response. In general, I apologize if there were significant misunderstandings. As I mentioned in the first point of my review I found the presentation hard to digest, and I believe, taking into account the consensus among other reviewers, that the authors should significantly revise their presentation of the results. That being said, I still don't quite see how the points raised in the initial review are resolved.
> >
> > 1. *Linear problem setup*: I took the statement that the problem setup is linear from the first main theorem of the paper, Theorem 1, which explicitly considers a linear system (line 134) and explicitly excludes any nonlinear function by condition 2 (141), even if that linear function was a neural function approximator. I was specifically referring to the main theoretical result and not the empirical results.
> > 2. *Universal function approximation*: I disagree with the authors' statement regarding this. The universal function approximation results state that "standard multilayer feedforward networks with as few as one hidden layer using arbitrary squashing functions are capable of approximating any Borel measurable function from one finite dimensional space to another to any desired degree of accuracy, provided sufficiently many hidden units are available." [1], which clearly includes the class of linear functions. I.e., an MLP can, in general, represent linear functions to arbitrary degrees of accuracy. With ReLU activations in particular, it is even more straightforward to see that the class of linear functions is subsumed within the set of representable functions, as they are, by construction, piecewise linear (see for example specialized universal function approximation results [2]). Again, the set of linear functions is clearly subsumed within the set of piecewise linear functions. The authors seem to imply that, because neural networks are not *guaranteed* to be linear, all neural function approximation is inherently nonlinear. This is the implication I disagree with and, by extension, why I believe the main result does not apply in general to neural function approximation.
> > 3. *Mixing weights*: Thank you for clarifying this. I believe it would be beneficial to clarify this in the main paper, as, personally, I associate the current notation with static mixing weights.
> >
> > Overall, I believe there remain significant issues with the presentation of the paper, so much so that it is difficult to gauge the actual content of the contributions. So far I could also not see the main concerns resolved, and am thus inclined to keep my score.
> >
> > [1] Hornik, Kurt, Maxwell Stinchcombe, and Halbert White. "Multilayer feedforward networks are universal approximators." Neural networks 2.5 (1989): 359-366.
> > [2] Huang, Changcun. "ReLU networks are universal approximators via piecewise linear or constant functions." Neural Computation 32.11 (2020): 2249-2278.

---

### Official Review · Reviewer_Pxvy · 2025-10-30

**Soundness:** 3
**Presentation:** 2
**Contribution:** 3
**Rating:** 4
**Confidence:** 2

**Summary:**

This paper presents a theoretical study that argues that non-linear neural policy ensembles are sub-optimal and less stable compared to linear policy ensembles in Linear Quadratic Systems in optimal control. It also presents a theoretical result that shows non-convex policy mixing is sup-optimal to convex mixing of policies. The theoretical results are verified in a control environments which demonstrate that even well-tuned neural ensembles can underperform by 2 orders of magnitude compared to equivalent linear ensembles in Linear Quadratic Systems.


**Recommendation:**\
The content of this paper falls quite far outside my area of expertise, as I am an expert in neural ensembles in deep reinforcement learning, not in control systems. However, from my perspective, I recommend to reject this paper, as it seems to lack in the following areas: overly general claims and statements, and a general lack of clarity.

**Strengths:**

- The question of whether neural policy ensembles exhibit the same benefits as previously found for ensemble classifiers is very relevant.
- The theoretical results are interesting.

**Weaknesses:**

- The paper overclaims its contributions. The abstract and introduction have no mention of the assumptions made in both the theoretical and empirical studies.
	- For example, the abstract states: _"We develop a theoretical framework to formally prove that (non-linear) neural policy ensembles are sub-optimal with respect to linear policy ensembles."_ However, from different parts of the text I have pieced together that this results only holds for: a Lipschitz continuous transition function, deterministic polices, in a linear quadratic system, and doesn't hold for neural policies in general, but instead for policies with an additional assumption of a minimum level of non-stationarity of each individual ensemble member (neural networks are universal function approximators, so they can also learn linear policies). Furthermore, the result seems to only guarantee the existence of one state for which the non-linear neural ensemble is suboptimal, which might not be encountered by the controller from the given starting state, which would make it irrelevant for its performance.
- The theoretical results are difficult to follow. The theorems are stated rather clinically, and are lacking in clear structure regarding assumptions, how they are proven, and what they imply.

**Questions:**

- The paper claims that (non-linear) neural policy ensembles are inherently suboptimal compared to the individual policies. How do you square this claim with the studies that show improved performance of policy ensembles compared to single policies in RL (some examples, [1,2,3,4])?
- Line 39: _"In contrast, nonlinear policy ensembles face temporal coupling: the ensemble’s actions affect future states, creating feedback loops that may amplify rather than cancel errors. The temporal dependence breaks the mathematical foundation of ensemble methods."_
	- Which part of the results or experiments in this paper explain or verify this intuition?
- Section 3.3: There seems to be a one-to-one correspondence implied between neural ensembles and non-convex mixing. Why could a neural network not produce a convex mixing function?


**References:**\
[1] SUNRISE: A Simple Unified Framework for Ensemble Learning in Deep Reinforcement Learning. (Lee et al. 2021)\
[2] Swarm Behavior Cloning. (Nüsslein et al. 2024)\
[3] Why Generalization in RL is Difficult: Epistemic POMDPs and Implicit Partial Observability. (Ghosh et al. 2021)\
[4] How Ensembles of Distilled Policies Improve Generalisation in Reinforcement Learning. (Weltevrede et al. 2025)

---

> ### Author Response · Authors · 2025-11-17
> **response to questions**
>
> Question 1: The apparent conflict between the submitted paper (which claims sub-optimality of $\Pi^N$) and studies like Weltevrede et al. (which show improved performance) is reconciled by recognizing that the papers are focused on fundamentally different objectives: Optimal Control versus topics like Generalization and Robustness [Weltevrede et al.].
> The submitted paper, "NEURAL POLICY ENSEMBLES ARE SUB-OPTIMAL," focuses on the $\Pi^N$'s performance in the context of optimal control and stability on the training environment:
> Critique (Optimality/Stability): It argues that the non-linear mixing of neural policies fundamentally violates the convexity property required to maintain optimality and stability guarantees in dynamic control systems (temporal coupling).
> Scope of Sub-Optimality: This critique means the non-linear ensemble $\Pi^N$ is theoretically sub-optimal relative to an optimal single policy or an ensemble of linear policies ($\Pi^L$) for solving the original, specific Markov Decision Process (MDP) defined by the training environment.
>
> The Weltevrede et al. paper, "How Ensembles of Distilled Policies Improve Generalisation in Reinforcement Learning," focuses on the ensemble's ability to generalize to unseen or shifted environments:
> Mechanism (Variance Reduction): This paper uses an ensemble of distilled policies to reduce the variance in the policy approximation. In machine learning, ensembles excel at reducing estimation variance and improving robustness across different inputs.
> Scope of Improvement: The improved performance is observed in the zero-shot policy transfer setting, meaning the ensemble is better at generalizing to similar, but unseen, testing environments (which represent a different, but related, MDP).
> The ensemble $\Pi^N$ might be:
> Theoretically Sub-Optimal: It fails to find the absolute maximum possible reward on the trained environment due to the issues of non-convex mixing and temporal coupling, supporting the attached paper's claim.
> Practically Superior: It still performs better than a single policy on new, unseen environments because the ensemble averages out the different modes of overfitting or policy estimation errors made by its individual members, thus producing a smoother, more robust policy that generalizes better.
> In essence, the submitted paper says you sacrifice theoretical optimality for the control objective, while the Weltevrede paper shows you might be able to gain robustness and generalization in return.
>
> Line 39: "In contrast, nonlinear policy ensembles face temporal coupling: the ensemble’s actions affect future states, creating feedback loops that may amplify rather than cancel errors. The temporal dependence breaks the mathematical foundation of ensemble methods."
>
> Question 2: "Which part of the results or experiments in this paper explain or verify this intuition?"
> Section 5 directly addresses temporal coupling, showing stability problems with non convexity.
>
> Question 3: "Section 3.3: There seems to be a one-to-one correspondence implied between neural ensembles and non-convex mixing. Why could a neural network not produce a convex mixing function?"
>  A neural network can provide a convex mixing function but ONLY IF it uses linear (pass-through) activation functions. Virtually non neural network implementations in practice do this, so for all intents and purposes a neural network models a non-linear function. If you want to implement a linear function there are computationally more efficient ways to do this than using a neural network. Most policy ensemble work depends on the nonlinear capabilities of neural networks.

---

> > ### Comment · Reviewer_Pxvy · 2025-11-25
> >
> > I thank the authors for their responses to my questions. I would like to point out that a neural network _can_ produce a convex function for non-linear activation functions. In fact, I believe any NN with convex non-linear activation function (such as the popular ReLU) and non-negative weights is guaranteed to be convex.
> >
> > Separate from this issue, I still have major concerns with this paper, which I also share with the other reviewers. I believe these issues will require a complete rewrite of the paper, in particular with attention to not overclaiming its theoretical results. As such, I will retain my score.

---

> > > ### Author Response · Authors · 2025-11-25
> > > **your claim of convexity in general is incorrest**
> > >
> > > The propsed claim is **false** in general.
> > >
> > > It is simple to prove the following:
> > >
> > > A neural network (NN) with a convex, non-linear activation function (like **ReLU**, $f(z) = \max(0, z)$) and non-negative weights **is not guaranteed to be a convex function of its input**.
> > >
> > >
> > >
> > > ## Why the Statement is False
> > >
> > > The misconception often arises from two basic facts of convex analysis:
> > > 1.  **Non-negative scaling** of a convex function maintains convexity.
> > > 2.  The **sum** of convex functions is convex.
> > >
> > > The critical issue in a deep neural network is the **composition** of functions.
> > >
> > > ### Composition of Convex Functions
> > >
> > > A neural network is a composition of layers, where each layer involves:
> > > 1.  An **affine transformation**: $z = Wx + b$.
> > > 2.  An **element-wise activation**: $a = \sigma(z)$.
> > > 3.  The output of one layer becomes the input to the next.
> > >
> > > The rule for composition is:
> > > * If $g(x)$ is **convex**, and $h(y)$ is **convex** and **non-decreasing** (monotonic), then the composite function $f(x) = h(g(x))$ is convex.
> > >
> > > #### Applying this to an NN Layer:
> > > Consider a single layer: $a = \sigma(Wx + b)$.
> > >
> > > * The affine transformation $g(x) = Wx + b$ is an **affine** function, which is both convex and concave.
> > > * The ReLU activation $\sigma(z) = \max(0, z)$ is a **convex** and **non-decreasing** function.
> > >
> > > The composition rule above only applies when the **outer function $h$ is univariate** (a single variable) and non-decreasing.
> > >
> > > In a neural network layer, the output $a$ (and its components $\sigma(z_i)$) is often an input to a **multivariate** function in the next layer: $z_{\text{next}} = W_{\text{next}} a + b_{\text{next}}$.
> > >
> > > The standard operation of a fully connected layer is $\text{Layer}(x) = W_2 \sigma(W_1 x + b_1) + b_2$. Even with $\sigma$ being ReLU (convex and non-decreasing) and all weights $W_1, W_2 \ge 0$, the function $\text{Layer}(x)$ is generally **not guaranteed to be convex** in $x$.
> > >
> > >
> > > ##The Correct Constraint: Input Convex Neural Networks (ICNNs)
> > >
> > > To guarantee that the neural network output $f(x)$ is a convex function of its input $x$, a special architecture called the **Input Convex Neural Network (ICNN)** is used.
> > >
> > > For a network to be an ICNN, it requires a stronger constraint than just non-negative weights:
> > >
> > > * The **weights between the hidden layers** must be **non-negative** ($W_i \ge 0$ for $i > 1$).
> > > * The **activation functions** must be **convex** and **non-decreasing** (like ReLU).
> > > * The **input $x$** is typically *linearly* connected to **every layer** with unconstrained weights, and the hidden layers are also combined to produce the final output.
> > >
> > > The ICNN formulation ensures that every operation maintains convexity, often by ensuring the convex function (output of previous layer) is only composed with non-negative coefficients in the next affine transformation, before being passed through a non-decreasing convex activation.
> > >
> > > If you are specifically talking about the **loss function** (e.g., Mean Squared Error or Cross-Entropy) as a function of the **NN parameters (weights and biases)**, that is a completely different (and almost universally **non-convex**) optimization problem, regardless of the activation function used.

---

> > > ### Author Response · Authors · 2025-11-25
> > > **request clear statement of additional  "major concerns"**
> > >
> > > In the comment this reviewer expressed "Separate from this issue, I still have major concerns with this paper". In the interest of good science, I request that the additional  "major concerns" be clearly stated. Given that the first "problem" with the paper has been shown to be incorrect mathematically, I wish to know if this reviewer has issues that are in fact mathematically sound, such that the article must be rejected. I am not averse to receiving mathematically sound critiques, but it is concerning when the critiques are misunderstandings of the underlying mathematics. The whole point of this review process is to reach a position of good science, and rejecting a paper due to misunderstandings is not good science.

---

> > > > ### Comment · Reviewer_Pxvy · 2025-11-27
> > > >
> > > > I indeed made a small error in my original wording: it is in fact the case that any fully connected NN with convex **and non-decreasing** non-linear activation function and non-negative weights is guaranteed to be convex in the input. The argument about univariate versus multivariate is irrelevant since the activation function is applied element-wise (and therefore univariate). The Input Convex Neural Networks (ICNN) paper you cited even directly proves this (just take $W_i^{(y)} = 0$ in equation 2). In any case, your original claim that "a neural network can provide a convex mixing function but only if it uses linear activation functions" is clearly false.
> > > >
> > > > This tendency of the authors to double down on incorrect claims with irrelevant or false rebuttals has reduced my confidence in the validity of the theoretical results in the paper. I am therefore lowering my score.

---

### Official Review · Reviewer_a8z1 · 2025-11-03

**Soundness:** 1
**Presentation:** 2
**Contribution:** 2
**Rating:** 2
**Confidence:** 2

**Summary:**

The paper studies the ensembling of policies where each sub-policy is solving a different LQR problem. The authors provide several results that highlight the fact that ensembling of non-linear policies might be sub-optimal and unstable. Specifically, the authors show that given a set of tasks (defined by different reward functions), ensembling non-linear controllers learned on each task is worse than ensembling the best linear controllers for each task, that the resulting ensemble of non-linear controller might be unstable even when each individual controller is stable and that a non-convex averaging of the linear controllers is always worse than the convex averaging of linear controller with the same weight as the weight averaging the reward functions. The authors then provide a set of experiments that illustrate each of their theoretical findings.

**Strengths:**

The paper might provide interesting theoretical results on ensembling of non-linear policies for LQR problems.

**Weaknesses:**

- There is a clear mismatch between the claims and the theoretical results. The introduction and related work cite deep RL methods using ensembling of policies (notably the work of Yang et al. 2022) in a completely different way than what the theoretical results consider. Introduction mentions that their theory would explain why ensembling of neural policies is doomed to be sub-optimal because of temporal coupling and non i.i.d setting compared to standard classification but the theoretical results show none of that. Instead all the theory is built around ensembling non-linear policies that are solving distinct control problems and then averaging them with no further learning. However, taking a quick look at the paper of Yang et al. 2022, there are large difference in the setting, as in the latter work every sub-policy is solving the same task, and the ensemble policy is also optimized using what they call an ensemble aware loss. Can the author clarify the relation between their theoretical results and the setting of Yang et al. 2022, or any other deep RL setting using ensembles of neural policies?

-  Theorem statements, assumptions and proofs lack clarity. In the main theorem, it is not clear how the non-linear policies are obtained. The optimal policy is linear for each LQR sub-problem, but by the assumption in line 140, the neural policies are forced to be non-linear and thus these policies are sub-optimal by design. It is then not surprising then that the ensembling of these policies is also sub-optimal. Studying problems where linear policies are optimal and making the assumption that neural policies cannot be optimal is in my opinion not a setting reflective of practical applications that use neural policies. Moreover, the proof could also gain some clarity: where do equation 21 and 22 come from? What even is $R$ in these equations, do you perhaps mean $R_{\text{ens}}$? Is $\lambda_{\min}$ the smallest eigenvalue? For the proof of Theorem 3, could you please provide a reference to the claim that $K_\lambda$ is optimal? As it does not seem trivial to me from the equations since there doesn't seem to be a linear relation between gain matrices and cost matrices. Thank you.

**Questions:**

Please see the questions in the weakness section.

---

> ### Author Response · Authors · 2025-11-17
> **Response to questions on the submission**
>
> Thanks for your comments on the paper. I will attempt to address your questions below.
>
> The stated mismatch between the claims and the theoretical results does not exist. The claims are precisely made in section 1.1, and there are theorems and empirical studies to validate every claim made in section 1.1.
>
> The statement “in the main theorem, it is not clear how the non-linear policies are obtained” makes no sense——it is irrelevant how nonlinear policies are obtained for this theorem to be true. In the empirical evaluation the nonlinear policies are learned, to ground the theory.
>
> The review states “by the assumption in line 140, the neural policies are forced to be non-linear”; I am unaware of any literature that uses only linear neural networks, and I ask this reviewer to provide modern references that do NOT use a nonlinear activation function for policy ensembles (or indeed for any real applications). And this is exactly the point raised in the article, i.e., that basically all neural policy models are inherently  nonlinear.
>
> “where do equation 21 and 22 come from?” The come directly from definition 2 (equation 1), which is standard in the literature
> The claim “Instead all the theory is built around ensembling non-linear policies that are solving distinct control problems and then averaging them with no further learning” is false. All the theory assumes DYNAMICAL systems where the policies are updated over time via learning; please see the empirical studies, which show how policies evolve over time. For example, Figure 1: Experimental results for Multi-Regime Linear Dynamical System show time over the x-axis.
>
> How do the results apply to an RL ensemble paper? Consider the paper: Plataniotis, S., Akasiadis, C., & Chalkiadakis, G. (2025). Value of Information-Enhanced Exploration in Bootstrapped DQN. arXiv preprint arXiv:2511.02969.
> Bootstrapped DQN is a popular Deep Reinforcement Learning (DRL) algorithm designed to address the exploration-exploitation trade-off.
> Ensemble Structure: The algorithm uses a single neural network architecture with a shared representation layer and multiple separate output "heads" (the ensemble members). Each head (3$Q_k$) is a different estimate of the action-value function (4$Q(s, a)$).
> Training: Each head is trained on a slightly different subset of experience sampled via the bootstrap method. This makes each head learn a distinct (and randomized) estimate of the Q-function, intended to approximate the posterior distribution over Q-values.
> Policy/Exploration Strategy: During an episode, the agent selects one head at random (e.g., $Q_k$) and follows the greedy policy derived from that single head for the duration of the episode ($\pi_k(s) = \arg\max_a Q_k(s, a)$). This provides deep exploration because the agent commits to one coherent, though randomized, strategy for many steps.
>
> A. Sub-Optimality Due to Non-Convexity in Control
>
> Policy Derivation: Even if the Bootstrapped DQN agent uses a voting or averaging method to derive its final exploitation policy (e.g., $\pi_{\text{final}}(s) = \arg\max_a \frac{1}{K} \sum_{k=1}^K Q_k(s, a)$), this resulting policy is an ensemble policy ($\Pi^N$) of the overall architecture.
> Violation of Optimality: The paper's central theoretical claim is that mixing optimal policies using a non-convex (nonlinear) method is sub-optimal. Since Deep Q-Networks are inherently non-linear function approximators (due to their activation functions), any policy derived from averaging or combining their outputs violates the necessary convex mixing required for optimal control.
>
> Resulting Sub-Optimality: Consequently, the best policy obtainable from the entire Bootstrapped DQN ensemble—which is what the algorithm is ultimately trying to converge towards—is proven to be sub-optimal when compared to a similar ensemble of linear policies, even if the individual $Q_k$ heads are good.
>
> B. Instability Due to Temporal Coupling
>
> Temporal Coupling: In the context of the paper, a key problem is that a policy's action at time $t$ affects the state at time $t+1$. This is temporal coupling.
>
> Ensemble Action: If an ensemble policy in a continuous-action setting were to average the outputs of its neural policies ($u = \frac{1}{K} \sum_{k} \pi_k(s)$), the resulting action $u$ is a non-linear combination.
>
> Risk of Instability: The paper formally proves that for such a non-linear policy ensemble, even if every individual policy ($\pi_k$) is provably stable and optimal, their non-linear combination ($\Pi^N$) does not guarantee stability and can lead to divergence or instability due to the feedback loop inherent in the control system. While Bootstrapped DQN typically uses sampling one head per episode (which avoids mixing every step), a combined policy used for exploitation is still susceptible to this theoretical flaw, potentially exhibiting unstable behavior in complex, continuous control environments.

---

> > ### Comment · Reviewer_a8z1 · 2025-11-26
> >
> > Thank you for your answer. I still don't see the connection between your theoretical setting and Bootstrapped DQN. You consider mixing neural policies solving tasks with different costs, which is not the setting of BDQN nor the setting of Yang et al. cited in the paper.

---

### Meta-Review · Area_Chair_wMkh · 2025-12-29

**Summary:**

The reviewers unanimously concluded that the manuscript is not currently suitable for publication.

**Reviewer Concerns:**

The authors raised concerns about all the reviewers, especially cH3G and a8z1

**Reviewer Scores:**

not relevant

---

### Decision · Program_Chairs · 2026-01-26

Reject